# Online Calibration of Inertial Sensors Based on Error Backpropagation

**DOI:** 10.3390/s24237525

**Published:** 2024-11-25

**Authors:** Vojtech Simak, Jan Andel, Dusan Nemec, Juraj Kekelak

**Affiliations:** Department of Control and Information Systems, Faculty of Electrical Engineering and Information Technology, University of Žilina, 010 26 Žilina, Slovakia; vojtech.simak@uniza.sk (V.S.); andel.jano@gmail.com (J.A.); juraj.kekelak@feit.uniza.sk (J.K.)

**Keywords:** global satellite navigation, inertial sensors, online calibration, error backpropagation

## Abstract

Global satellite navigation systems (GNSSs) are the most-used technology for the localization of vehicles in the outdoor environment, but in the case of a densely built-up area or during passage through a tunnel, the satellite signal is not available or has poor quality. Inertial navigation systems (INSs) allow localization dead reckoning, but they have an integration error that grows over time. Inexpensive inertial measurement units (IMUs) are subject to thermal-dependent error and must be calibrated almost continuously. This article proposes a novel method of online (continuous) calibration of inertial sensors with the aid of the data from the GNSS receiver during the vehicle’s route. We performed data fusion using an extended Kalman filter (EKF) and calibrated the input sensors through error backpropagation. The algorithm thus calibrates the INS sensors while the GNSS receiver signal is good, and after a GNSS failure, for example in tunnels, the position is predicted only by low-cost inertial sensors. Such an approach significantly improved the localization precision in comparison with offline calibrated inertial localization with the same sensors.

## 1. Introduction

The localization of mobile objects is an integral part of modern life. Global navigation satellite systems (GNSSs) [1] are most often used for localization in the outdoor environment. However, they have their limitations, especially due to bad weather conditions, densely built-up areas (e.g., larger cities), or when passing through tunnels. Therefore, mobile object localization systems need to be a combination of multiple localization sources. A common approach is to combine GNSSs with an inertial measurement unit (IMU), which contains inertial sensors (accelerometer, gyroscope), and, optionally, other sensors (e.g., magnetic compass). Navigation-grade inertial navigation systems (INSs) are usually very expensive since they contain optical gyroscopes due to their high accuracy. Another possibility is the use of LIDAR technology [2] or cameras [3]. For localization in buildings or halls, GNSS localization may be combined with localization based on the measurement of relative signal strength (RSSI) between multiple fixed transmitters and a mobile receiver, or vice versa [4,5]. The most frequently used wireless technologies are BLE, Wi-Fi, or proprietary UVW (Ultra-Wide Band) protocols. The development and rapid deployment of low-cost MEMS IMU units that are integrated into wearable electronics opens new possibilities for more accurate position prediction, even for small and affordable devices. However, the biggest disadvantage is the inaccuracy of MEMS IMU sensors, whose errors (noise, bias) are greater than that of optical IMUs, and it is necessary to calibrate them almost continuously.

When combining different sensors, we must solve the problem of sensor fusion. The Kalman filter or complementary filter is most often used for the fusion [6]. Many articles have discussed the fusion of readings from GNSSs and IMUs that use traditional mathematical methods based on complementary filters and Kalman filters [6,7,8,9,10,11,12]. The EKF and its derivatives is the de facto standard approach for sensor fusion, but there are a few disadvantages as follows:The model of the system needs to be known (including its parameters).The EKF assumes the sensors’ noise is normally distributed.The vanilla version of the EKF assumes the noise covariance matrices are constant and a priori known. For time-variant systems, this issue can be addressed by an adaptive EKF, which estimates the covariance of inputs from previous samples; see, e.g., [13,14].The vanilla EKF does not handle sensor bias well. The augmented EKF appends unknown parameters to the state vector (seem e.g., [15]); hence, it can be used to estimate and compensate for input bias.For highly non-linear systems, the sensor noise needs to be reasonably low since the EKF is a first-order linear approximation. Otherwise, the filter becomes unstable. This issue can be partially addressed by the error state extended Kalman filter (ES-EKF), which does not estimate the state directly but estimates only the correction of the state with respect to the incrementally predicted value, which can be less non-linear [16].

In real-world scenarios, the above conditions are usually not met. Especially, in continuous long-term applications, the noise characteristics of input sensors and the parameters of the system may change over time due to degradation, malfunction, and external stimuli. Especially, changes in the sensors’ calibration constants have a significant negative impact on the overall performance of the localization system.

We may also perform data fusion through advanced methods using artificial intelligence (AI) [17,18]. There are many approaches in which we can combine traditional methods with AI. In such an approach, the adaptive extended Kalman filter (AEKF) is most often used for data fusion, which is adapted based on the output of a trained neural network [19]. However, we can also use neural networks themselves for data fusion. The use of recurrent neural networks, specifically gated recurrent units (GRUs) and long short-term memory units (LSTMs) [20], has proven to be the most effective among the neural-network-based data fusion techniques. There are also approaches via traditional convolution neural networks (CNNs), which proved to be effective only for determining the direction of rotation, but the input was uncalibrated data [21]. The main drawback of AI-based sensor fusion is its low predictability; hence, we cannot guarantee the behavior of the system for all possible inputs. The main challenge of sensor fusion is the estimation of quality for each individual sensor subsystem and the optimal combination of their readings by decreasing the influence of low-quality sources in favor of higher-quality sources. The first GNSS was a GPS, which reached tens of meters of accuracy. With the deployment of other navigation satellite systems (GLONASS, Galileo, BeiDou) and their supporting SBAS systems, accuracy has increased, especially with modern dual-band GNSS receivers. GNSS precision can be further improved by multiple approaches, such as troposphere estimation [22], obtaining the clock offset [23], or we may use a distributed localization system containing several GNSS receivers combined using a linear Kalman filter (KF) [24].

Before the data fusion itself, it is necessary to eliminate the systematic errors (gain/bias deviations) of the input, commonly by the calibration of the input sensors. The Zero Velocity Potential Update (ZUPT) algorithm is most often used to calibrate the gyroscope, which calibrates the gyroscope offset whenever the IMU is motionless [25]. Such an algorithm is used especially in wearable electronics where the device spends a lot of time in idle mode. To calibrate the accelerometer, turning the object to multiple sides is often used, but this approach is not always possible and cannot be performed continuously. There are several approaches to calibrating a magnetometer. The most common is through the search for an ellipsoid. We can also use this approach in the case of an accelerometer [25,26,27]. Some approaches solve the ellipsoid problem via maximum likelihood (ML) estimation [28]. Some calibration algorithms estimate also the misalignment between the axes of different sensors [29,30]. To estimate the attitude (rotation) of the vehicle, we may use multichannel phase-locked GNSS receivers [31,32].

There are several possible approaches to calibration and, subsequently, to data fusion, and each of them increases the accuracy of the localization of mobile objects, especially in the event of a GNSS communication failure. The best approach is probably to use a combination of several approaches. It is advantageous to use the EKF for data fusion, but its adaptation should be solved through a recurrent neural network that looks for anomalies in the data. The calibration of the sensors should be solved through the knowledge of movement with a good signal from GNSSs. A large amount of data is needed to train a recurrent network that would adapt the EKF. We focused on the calibration of the sensors based on the knowledge of the motion vector through AI tools. We designed and built a module for logging data from the vehicle, and we subsequently processed and determined the accuracy of position prediction through various approaches of sensor fusion and calibration.

## 2. Data Logger

We designed and built a module for logging data from a passenger vehicle. The module has several sensors, specifically, a MEMS IMU unit, a barometer, and a GNSS receiver. The main microcontroller is ESP32, which has a Wi-Fi and Bluetooth wireless connection. It detects information from the car’s control unit via the ODBII connector via Bluetooth wireless communication. The module logs all data on the SD card to a CSV file. After arriving at a certain location, it connects to the Wi-Fi network and sends the measured data to the server for further processing. The module was described in more detail in [33,34]. The ground-truth location has been captured by SPAN-CPT from NovAtel, which is a navigation-grade GPS+IMU system incorporating fiber optic gyroscopes (FOGs) and MEMS accelerometers, which are tightly coupled with a GNSS receiver. We placed the data logger module on the dashboard in a 3D-printed box, aligned with the axes of the vehicle (Figure 1). The location on the dashboard is not optimal since there is the structure of the vehicle under the module, which strongly deforms Earth’s magnetic field near the logger. On the other hand, this setup is very close to the real situation in the vehicle when the user mounts the navigation hardware on the dashboard or windshield. We placed the SPAN-CPT system above the rear axle of the vehicle and configured it for our purposes. The GNSS receiver antenna for the SPAN-CPT system was placed on the roof rails of the passenger car (see Figure 2).

We logged data during normal routes of a passenger vehicle and also routes directly determined for the purpose of data logging. We have laid out a route in which there are underground objects, such as garages and highway tunnels, in which a failure of GNSS communication is guaranteed. In total, we completed 12 trials, from which we had enough data to analyze several algorithms.

## 3. Data Fusion Through AEKF

When navigating in 3D space, it is important to set the coordinate system in which the location of the vehicle is determined. In GNSS localization, each position is represented by latitude, longitude, and altitude, which is the height above the mean sea level (MSL). However, when using INSs, a local world Cartesian (orthonormal) x, y, and z coordinate system oriented tangentially to the normalized Earth’s surface is used. The origin (0, 0, 0) of the local world coordinate system corresponds to the initial position of the vehicle, and the curvature of the Earth is neglected. We can use this approach only in the case when mobile objects move at a low speed, or for a short time; hence, the traveled distance is negligible with respect to the Earth’s curvature radius. The body’s frame of reference is oriented according to NED convention (*x*—forward, *y*—right, *z*—down).

Single GNSS receiver estimates only the position but not the attitude (rotation) of the vehicle. If the vehicle is moving, we can estimate the rotation around the *z*-axis (yaw-*γ*) and the body’s *y*-axis (pitch-*β*) from the GNSS data, while the body’s *x*-axis (Roll-*α*) rotation is unknown. INS systems use gyroscopes and accelerometers, hence providing the measurement of angular velocity and linear acceleration in all three axes of the vehicle’s body. 

The gyroscope measures rotational acceleration in individual axes. It does not matter where it is located on the mobile object, as the rotation of the object is the same everywhere. We can calculate the changes of Euler angles *α*, *β*, *γ* from the angular velocity vector [ω_x_, *ω_y_*, *ω_z_*] measured by a gyroscope using the transformation (1). This approach is suitable, especially for small angles and a short time ∆*t* [35].
(1)$$\left[\begin{array}{c} \Delta \alpha \\ \Delta \beta \\ \Delta \gamma \end{array}\right]=\left[\begin{array}{ccc} 1 & \frac{sin(\alpha )sin(\beta )}{cos(\beta )} & \frac{cos(\alpha )sin(\beta )}{cos(\beta )}\\ 0 & cos(\alpha ) & -sin(\alpha )\\ 0 & \frac{sin(\alpha )}{cos(\beta )} & \frac{cos(\alpha )}{cos(\beta )} \end{array}\right]\cdot \left[\begin{array}{c} \omega_{x}\\ \omega_{y}\\ \omega_{z} \end{array}\right]∆t$$

If the sensor unit is not accelerating, it only measures gravitational acceleration and noise, which can determine the direction of the center of the Earth. After subtracting the gravitational acceleration from the readings of the accelerometer and subsequent integration, we obtain the speed of the object; and after integrating the angular velocity measured by gyroscope, we obtain a rotation in individual axes. From the given values, we can calculate the direction vector, which, after integration, creates the traveled path. Theoretically, we may read the value of the gravitational acceleration from the accelerometer data, but we must know the attitude of the mobile object and its linear acceleration. The measured value of the accelerometer is expressed by relation (2), where ***g*** is the gravitational acceleration, ***a***_0_ is the acceleration of the vehicle, and ***a***_n_ is the noise, which also includes the vibrations of the vehicle. All vectors are measured in the local coordinates of the vehicle. Due to the large noise and the strong effect of gravitational acceleration, the accelerometer is usually used to detect downward direction, hence estimating the roll and pitch of the vehicle. The rotation around the vertical (z) axis does not affect accelerometer; hence, the yaw angle cannot be determined.
(2)$$a=-g+a_{0}+a_{n}$$

The magnetometer measures the Earth’s magnetic field, which, however, is affected by the surrounding environment, so it is necessary to calibrate the magnetometer. Knowing the location of a mobile object, we can determine the inclination and declination (Figure 3) in the area, thanks to which we can determine the geographical north from the estimated magnetic north direction. The magnetometer is not sensitive to the rotation of the object around the axis parallel to the magnetic vector. By fusing the data from the magnetometer and the accelerometer, we can more accurately obtain information about the rotation of the vehicle.

We can estimate the speed of a mobile object from an accelerometer, but, as mentioned, this method is highly inaccurate due to the vibrations of the vehicle. For a ground vehicle, it is more precise to obtain the speed through an odometer. We calculate the direction vector of the mobile object using Equations (3)–(5). First, we calculate the rotation matrix (***R***) from the Euler angles. Symbols *c_x_* and *s_x_* are shorthand for cos(*x*) and sin(*x*), respectively. The forward direction vector ***n_w_*** in world coordinates is simply the first column of the rotation matrix. Subsequently, we multiply the direction vector by the scalar speed of the mobile object (*v*). By the numerical integration of the velocity vector in the world coordinates, we obtain the position of the vehicle (***D***) (6).
(3)$$R=\left[\begin{array}{ccc} c_{\beta }c_{\gamma } & s_{\alpha }s_{\beta }c_{\gamma }-c_{\alpha }s_{\gamma } & c_{\alpha }s_{\beta }c_{\gamma }+s_{\alpha }s_{\gamma }\\ c_{\beta }s_{\gamma } & s_{\alpha }s_{\beta }s_{\gamma }+c_{\alpha }c_{\gamma } & c_{\alpha }s_{\beta }s_{\gamma }-s_{\alpha }c_{\gamma }\\ -s_{\beta } & s_{\alpha }c_{\beta } & c_{\alpha }c_{\beta } \end{array}\right]$$
(4)$$n_{w}=R\cdot \left[\begin{array}{c} 1\\ 0\\ 0 \end{array}\right]$$
(5)$$v_{w}=vn_{w}$$
(6)$$D=\sum v_{w}∆t$$

The above equations assume that the IMU axes are aligned with the vehicle’s axes. The process of compensating for the misalignment of individual sensors will be described in the next section.

We may compensate for the altitude error (*z* coordinate) using a barometric sensor. It senses the atmospheric pressure, which is directly proportional to the altitude. The conversion between altitude and atmospheric pressure is carried out using the well-known formula as follows: (7)$$h^{bar}=\frac{RTln\left(\frac{P}{P_{0}}\right)}{-gm}$$
where *P*_0_ is the sea level pressure, *P* is the gauge pressure, *m* is the molecular weight, *g* is the gravitational acceleration, *R* is the universal gas constant, and *T* is the thermodynamic temperature. Some atmospheric pressure sensors directly output altitude, while the calculation is carried out directly by the sensor’s built-in CPU. Such a reading of the altitude contains a non-constant offset, as the atmospheric pressure at sea level varies over time.

We can use the EKF to fuse data from the IMU unit. Figure 4 shows the EKF block diagram where the input is data from the IMU unit and the output is Euler angles. During the prediction cycle, the data from the gyroscope are added according to (1). During the update process, based on the prediction, *α*, *β*, and *γ* are calculated, which should be the values from the accelerometer and magnetometer; they are subtracted from the real data, which creates an error vector.

Prediction phase of the EKF:(8)$${\hat{s}}_{k}=f(s_{k-1},\;u_{k})$$
(9)$${\left[\begin{array}{c} \alpha \\ \beta \\ \gamma \end{array}\right]}_{k+1}={\left[\begin{array}{c} \alpha \\ \beta \\ \gamma \end{array}\right]}_{k}+∆t\left[\begin{array}{ccc} 1 & \frac{s_{\alpha }s_{\beta }}{c_{\beta }} & \frac{c_{\alpha }s_{\beta }}{c_{\beta }}\\ 0 & c_{\alpha } & -s_{\alpha }\\ 0 & \frac{s_{\alpha }}{c_{\beta }} & \frac{c_{\alpha }}{c_{\beta }} \end{array}\right]\cdot \left[\begin{array}{c} \omega_{x}\\ \omega_{y}\\ \omega_{z} \end{array}\right]$$
(10)$${\hat{P}}_{k}=F_{k}P_{k-1}F_{k}^{T}+G_{k}U_{k}G_{k}^{T}$$
(11)$$F_{k}=\frac{\partial f(s_{k},u_{k})}{\partial s_{k}};\;G_{k}=\frac{\partial f(s_{k},u_{k})}{\partial u_{k}}$$

The input ***u***_k_ of the prediction cycle is the data from the gyroscope, the angular velocity vector, and the state ***s***_k_ of the filter contains all Euler angles. Matrices *F* and *G* are partial derivatives of the function *f* in Formula (8). *P* is a 3 × 3 matrix representing the covariance of the state vector ***s***. The matrix *U* determines the variance of the input data of the gyroscope.

Updated phase of the EKF:(12)$$e_{k}=q_{k}-h({\hat{s}}_{k})$$
(13)$$\left[\begin{array}{c} {ea}_{x}\\ {ea}_{y}\\ \begin{array}{c} {ea}_{z}\\ {eM}_{x}\\ \begin{array}{c} {eM}_{y}\\ {eM}_{z} \end{array} \end{array} \end{array}\right]=\left[\begin{array}{c} {qa}_{x}\\ {qa}_{y}\\ \begin{array}{c} {qa}_{z}\\ {qM}_{x}\\ \begin{array}{c} {qM}_{y}\\ {qM}_{z} \end{array} \end{array} \end{array}\right]-\left[\begin{array}{c} gs_{\beta }\\ gc_{\beta }s_{\alpha }\\ \begin{array}{c} gc_{\beta }c_{\alpha }\\ Mc_{\beta }c_{\delta }c_{\gamma }c_{\theta }-Ms_{\beta }s_{\theta }+Mc_{\beta }c_{\theta }s_{\delta }s_{\gamma }\\ \begin{array}{c} Mc_{\beta }s_{\alpha }s_{\theta }-Mc_{\delta }c_{\theta }(c_{\alpha }s_{\gamma }-c_{\gamma }s_{\alpha }s_{\beta })+Mc_{\theta }s_{\delta }(c_{\alpha }c_{\gamma }+s_{\alpha }s_{\beta }s_{\gamma })\\ Mc_{\alpha }c_{\beta }s_{\theta }+Mc_{\delta }c_{\theta }\left(s_{\alpha }s_{\gamma }+c_{\alpha }c_{\gamma }s_{\beta }\right)-Mc_{\theta }s_{\delta }(c_{\gamma }s_{\alpha }-c_{\alpha }s_{\beta }s_{\gamma }) \end{array} \end{array} \end{array}\right]$$
(14)$$S_{k}=H_{k}P_{k}H_{k}^{T}+R_{k}$$
(15)$$K_{k}=H_{k}P_{k}S_{k}^{-1}$$
(16)$$s_{k}={\hat{s}}_{k}+K_{k}e_{k}$$
(17)$$P_{k}=(I-K_{k}H_{k}){\hat{P}}_{k}$$
(18)$$H_{k}=\frac{\partial h(s_{k})}{\partial s_{k}}$$

During the update, the error vector ***e*** is first calculated, which is equal to the difference between the measured values from the accelerometer and magnetometer sensors and the calculated value from the prediction of the Euler angles from the knowledge of inclination-*θ* and declination-*δ*. The matrix *R* determines the noise variance of individual sensors. The *K* matrix determines the Kalman gain. The output of the update is the most probable value of the Euler angles from the IMU unit.

To predict the trajectory, we need to know the speed of the mobile object. We can obtain it from the odometer or accelerometer after subtracting the gravitational acceleration. We can use a linear Kalman filter for data synthesis. We assume that the mobile object and the accelerometer have aligned axes and that the mobile object can only accelerate in the x-axis direction. From the KF output, we can calculate the effect of gravitational acceleration in the direction of the x-axis. In prediction, the input is the acceleration sensor, and in the update, the error is corrected using velocity $v_{k}^{odo}$ obtained from odometer data. Figure 5 shows the block diagram of the described KF.

Prediction:(19)$${\hat{x}}_{k}=Ax_{k-1}+Bu_{k}$$
(20)$${\hat{v}}_{k}=1{\cdot v}_{k-1}+∆t\cdot (a_{x}-g_{x})$$
(21)$${\hat{P}}_{k}=A_{k}P_{k-1}A_{k}^{T}+Q_{k}=P_{k-1}+Q_{k}$$

Update:(22)$${\delta v}_{k}=v_{k}^{odo}-H\;\cdot {\hat{v}}_{k}$$
(23)$$K_{k}=\frac{{\hat{P}}_{k}H_{k}^{T}}{H_{k}{\hat{P}}_{k}H_{k}^{T}+R_{k}}$$
(24)$$v_{k}={\hat{v}}_{k}+K_{k}{\delta v}_{k}$$
(25)$$P_{k}=(I-K_{k}H_{k}){\hat{P}}_{k}$$

Such a KF is simple and easy to calculate, and all values are scalar. The *Q* value expresses the noise variance of the accelerometer and *R* the noise variance of the odometer. The output of the synthesis is the most likely speed of the mobile object. Knowing Euler angles and speed, we can calculate the direction vector through the already described Equations (3) and (5). By adding the direction vector, we obtain the vehicle’s traveled path (6). With this approach, we created the INS inertial navigation system.

We also used the KF for the fusion between the INS and GNSS. The system is linear, so we did not have to use the extended Kalman filter. We also added values from the barometer to the input. The output from the KF was the most likely position of the mobile object in global coordinates. The KF equations are shown in (26)–(33). Matrices *A* and *B* are unitary, so during prediction, data from the INS are only added to the currently predicted value. During the update process, an error vector ***e*** is calculated, in which there are four values, *e_x_*, *e_y_*, *e_z_*, and height correction *e_h_*. The Kalman gain matrix *K* will have the form 4 × 3.

Prediction:(26)$${\hat{s}}_{k}=A_{k}s_{k-1}+B_{k}u_{k}$$
(27)$${\left[\begin{array}{c} x\\ y\\ z \end{array}\right]}_{k}=\left[\begin{array}{ccc} 1 & 0 & 0\\ 0 & 1 & 0\\ 0 & 0 & 1 \end{array}\right]{\cdot \left[\begin{array}{c} x\\ y\\ z \end{array}\right]}_{k-1}+\left[\begin{array}{ccc} 1 & 0 & 0\\ 0 & 1 & 0\\ 0 & 0 & 1 \end{array}\right]\cdot \left[\begin{array}{c} {\Delta x}_{INS}\\ \Delta y_{INS}\\ {\Delta z}_{INS} \end{array}\right]$$
(28)$${\hat{P}}_{k}=A_{k}P_{k}A_{k}^{T}+Q_{k}$$

Update:(29)$$e_{k}=q_{k}-H\;\cdot {\hat{s}}_{k}$$
(30)$${\left[\begin{array}{c} e_{x}\\ e_{y}\\ \begin{array}{c} e_{z}\\ e_{h} \end{array} \end{array}\right]}_{k}=\left[\begin{array}{c} \begin{array}{c} x^{GNSS}\\ y^{GNSS} \end{array}\\ \begin{array}{c} z^{GNSS}\\ h^{bar} \end{array} \end{array}\right]-\left[\begin{array}{c} \begin{array}{ccc} 1 & 0 & 0 \end{array}\\ \begin{array}{ccc} 0 & 1 & 0 \end{array}\\ \begin{array}{c} \begin{array}{ccc} 0 & 0 & 1 \end{array}\\ \begin{array}{ccc} 0 & 0 & 1 \end{array} \end{array} \end{array}\right]\cdot \left[\begin{array}{c} x\\ y\\ z \end{array}\right]$$
(31)$$K_{k}=\frac{{\hat{P}}_{k}H_{k}^{T}}{H_{k}{\hat{P}}_{k}H_{k}^{T}+R_{k}}$$
(32)$$s_{k}={\hat{s}}_{k}+K_{k}e_{k}$$
(33)$$P_{k}=(I-K_{k}H_{k}){\hat{P}}_{k}$$

In the described way, we can determine the most likely position of the vehicle, but we must have knowledge of the noise variance of the sensors before running the algorithm, and all sensors must be calibrated. We can perform sensor calibration using traditional methods. However, our approach is to calibrate the sensors based on the position from the GNSS at a time when the GNSS signal is good. Calibration takes place through the backpropagation of the error through the entire algorithm. At the output of the sensors, we added neurons that serve as a calibration matrix, and their parameters are adjusted based on the backpropagation of the error. With such an approach, we guarantee a constant calibration of the sensors, and in the event of a GNSS failure, the position is predicted based on the INS. Kalman filter functions are written as classes under the “PyTorch” library, which uses tensors through which it can calculate the backpropagation of the error. Individual neural networks are placed between the sensors and the position prediction calculation. The neural network for the gyroscope, magnetometer, and accelerometer consists of three neurons (Figure 6), the inputs of which are connected to each neuron, and thanks to which, we can obtain dependencies between individual axes; for example, if the sensor is rotated not completely in the direction of travel. Such a calculation creates a 4 × 3 calibration matrix (34) and (35). For one neuron, we obtain only offset and gain. We set the initial parameters of individual neurons to *w*_nk_ = 1 and *b*_n_ = 0 so that the sensor readings are unaltered. This approach considers only first-order calibration. Using a more complex neural network, we could also compensate for certain non-linearity, but the training is more computationally demanding, and the calibration would take longer to converge. The block diagram of the described method is shown in Figure 7.
(34)$$o_{1}=b_{1}+\sum_{i=1}^{n}w_{1n}i_{n}$$
(35)$$\left[\begin{array}{c} o_{1}\\ o_{2}\\ o_{3} \end{array}\right]=\left[\begin{array}{cc} \begin{array}{cc} w_{11} & w_{21}\\ w_{12} & w_{22}\\ w_{13} & w_{23} \end{array} & \begin{array}{cc} w_{31} & b_{1}\\ w_{32} & b_{2}\\ w_{33} & b_{3} \end{array} \end{array}\right]\cdot \left[\begin{array}{c} i_{1}\\ i_{2}\\ \begin{array}{c} i_{3}\\ 1 \end{array} \end{array}\right]$$

The algorithm processes the data if the HDOP value from the GNSS receiver is lower than 10, which represents barely acceptable quality of a GNSS lock. The backpropagation calculation is applied through the loss function. Through optimization, new neuron constants are recalculated, symbolizing our calibration constants. An error vector is calculated between the predicted position and the value from the GNSS. We use the criterion for the loss error function “MSEloss”, which calculates the quadratic error. We use the Adam algorithm [36] as an optimization method with a learning rate of 0.0001. The algorithm changes the calibration constants during the runs. The longer we have a good signal from the GNSS, the more accurate the calibration constants the algorithm will find. 

## 4. Achieved Results and Discussion

We wanted to compare the evaluation of the achieved results against the SPAN-CPT system. In Figure 8, we can see a comparison of the vehicle speed during a GNSS signal failure. The algorithm adjusted the calibration constants for the accelerometer and odometer that entered the KF. In Figure 9, we can see the elevation comparison. During the GNSS outage, altitude was predicted based on INS and barometer data. We compared the data with the reference sample from the SPAN-CPT system. We displayed the resulting data from the prediction on maps and compared it to the position from SPAN-CPT. As we can see in Figure 10, position prediction from the SPAN-CPT system was not accurate, even though it contained high-quality inertial sensors.

Hence, the position prediction from SPAN-CPT dead reckoning was not accurate enough to be considered as a reference value during an outage. That is why we decided to compare the data against the route generated from the map documents. But, we ran into a problem, where the maps of the tunnels from several sources did not correspond to reality since they do not match with each other. Online maps may show only a rough estimate of the underground road trajectory. Each of the tested map documents had the trajectory of the tunnels drawn slightly differently. The most accurate was the maps provided by Google. We have compared Google maps with the official maps provided by the Slovak Road Administration agency [37]. Hence, we generated a trajectory along which we walked and compared the data. We created a function that calculates the tangent to the trajectory from map data. The trajectory is shown in Figure 10, and the summary statistics are in Table 1. However, the comparison is not completely accurate, as the generated path was not the real road along which the vehicle traveled; also, when passing certain right-angled turns, the trajectory was curved, while in the generated route, the turn was sharp without shortening the trajectory. We have compared the errors of the proposed neural-calibrated Kalman filter system with the same system, where the neural networks were replaced by standard calibration parameters (gain, bias). The system was assembled in the same way, but at the input, there were no neural networks but calibration matrices. We have calibrated the gyroscope through the ZUPT algorithm (bias calibration was performed whenever the vehicle was stationary). Calibration of the magnetometer was performed based on the knowledge of the traveled path [33]. We did not calibrate the accelerometer in place because it was not possible to rotate the object (passenger vehicle) in all directions.

The results in Figure 11 and Figure 12 show that calibration through error backpropagation is feasible and has better results than traditional methods, including the navigation grade GNSS + INS system. The proposed method takes advantage of the readings from the odometer; hence, it is primarily applicable to wheeled vehicles. Although data processing took place offline, the entire algorithm works in online mode (processing data sample by sample, without the need to know all future values), and it is possible to integrate it into the software of an onboard signal processing microcomputer. The main disadvantage is that the algorithm requires slightly more computing power compared to traditional approaches. Nowadays, however, this is not such a problem. The main advantage of the algorithm is its adaptability and continuous sensor calibration, thanks to which we can use low-cost sensors with lower accuracy and predict the position of the vehicle during GNSS outages more accurately than expensive systems.

## 5. Conclusions

More accurate localization of mobile objects is still an open topic. GNSS systems or their combination with INSs are most often used for localization. However, INS systems require precisely calibrated sensors for the calculation to be correct. There are several options for calibrating the sensors within INS systems. We focused on the possibilities of using AI tools for the task, while we proposed an algorithm that consists of three KFs estimating the position and attitude of the vehicle. Instead of sensor calibration constants, we connected neural networks (NNs) between the filter input and the sensors. The parameters of the NNs are adjusted using error backpropagation. Backpropagation always takes place when the GNSS signal is good (HDOP value is below a threshold). In this way, the algorithm can update the calibration constants of individual sensors and at the same time find dependencies between the axes, which calibrates the inaccuracies of the axis alignment. We verified the algorithm on our own measured data from a passenger vehicle and verified the values against the same algorithm via the KF but with calibration via traditional methods. At the same time, we also compared the prediction with the navigation-grade GNSS+INS system. We expected such a system to be a reference system, but it turned out that after the GNSS failure, the position prediction had an increasing error, which was significant. Therefore, we compared the position estimate against the route generated from the online map documents. The results show that our algorithm achieved the best results.

The result of this work is a clear conclusion that artificial intelligence tools can improve the prediction of the position of mobile objects, especially in combination with traditional mathematical procedures that solve the fusion between sensors. Artificial intelligence tools can then be used to calibrate the sensors, adapt the KF parameters, or detect anomalies in the data from the sensors and then remove them from the fusion process. Such approaches are especially suitable for affordable MEMS sensors, whose calibration requirement strongly affects the accuracy of position prediction.

Future work may focus on implementing the proposed approach to different types of vehicles (e.g., aerial drones) that cannot take advantage of the wheel odometry. It is also a viable option to test the ES-EKF (error-state EKF) instead of the EKF, which may improve the precision of the localization even further.

## Figures and Tables

**Figure 1 sensors-24-07525-f001:**
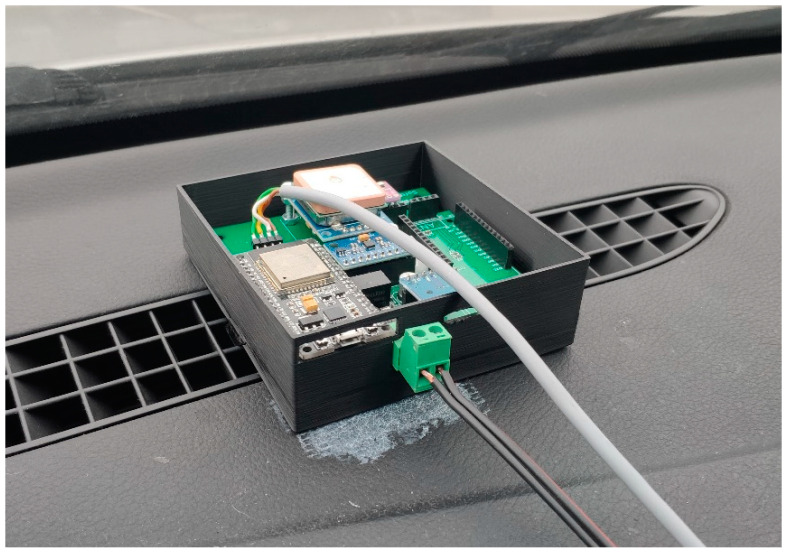
Integration of the module into a passenger vehicle.

**Figure 2 sensors-24-07525-f002:**
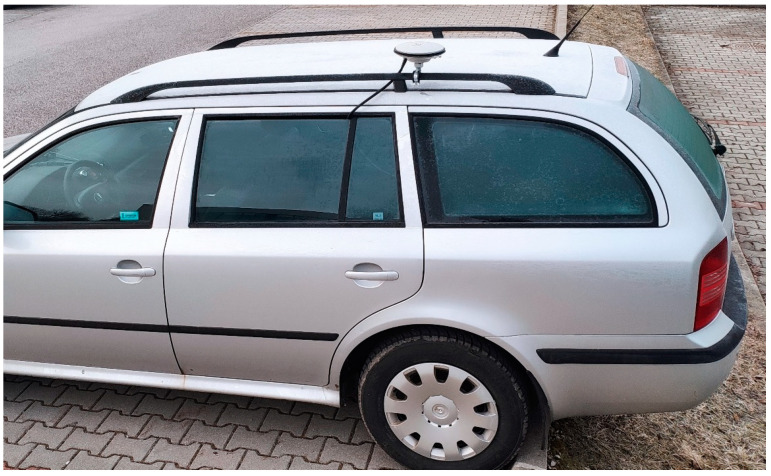
GNSS antenna placed on a passenger vehicle.

**Figure 3 sensors-24-07525-f003:**
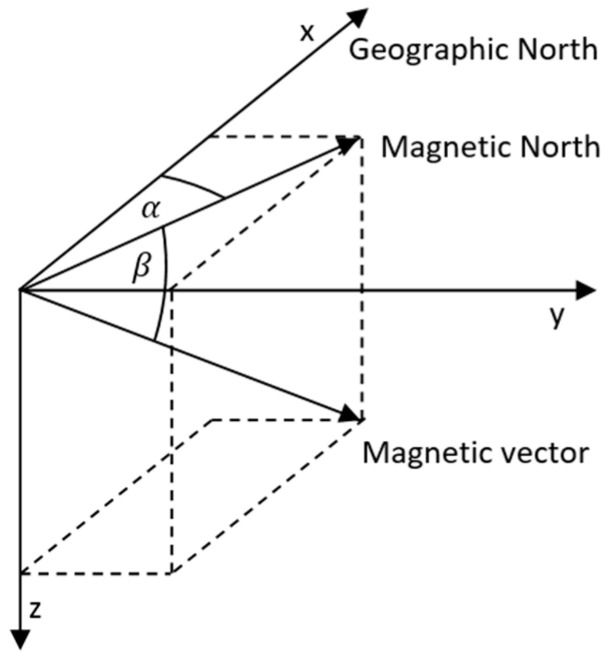
Inclination and declination of the magnetic field.

**Figure 4 sensors-24-07525-f004:**
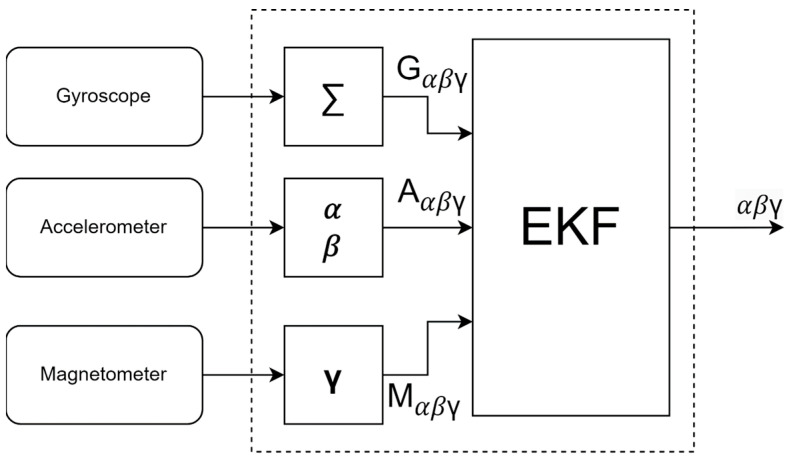
Extended Kalman filter for the IMU unit.

**Figure 5 sensors-24-07525-f005:**
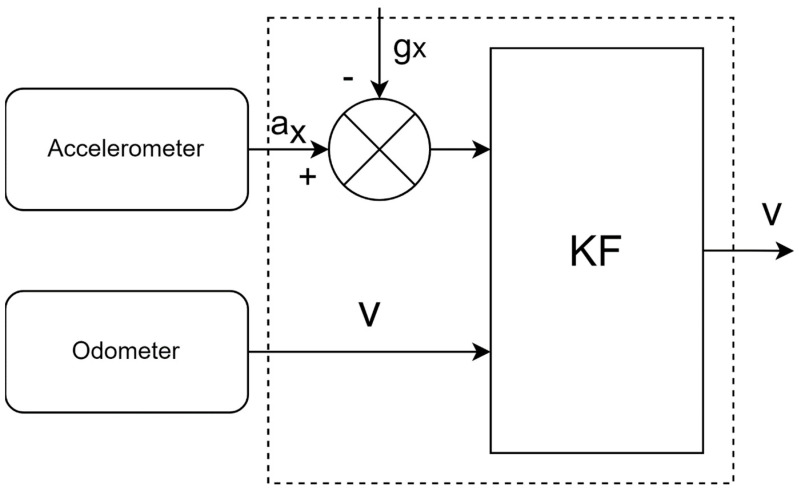
Kalman filter for velocity update.

**Figure 6 sensors-24-07525-f006:**
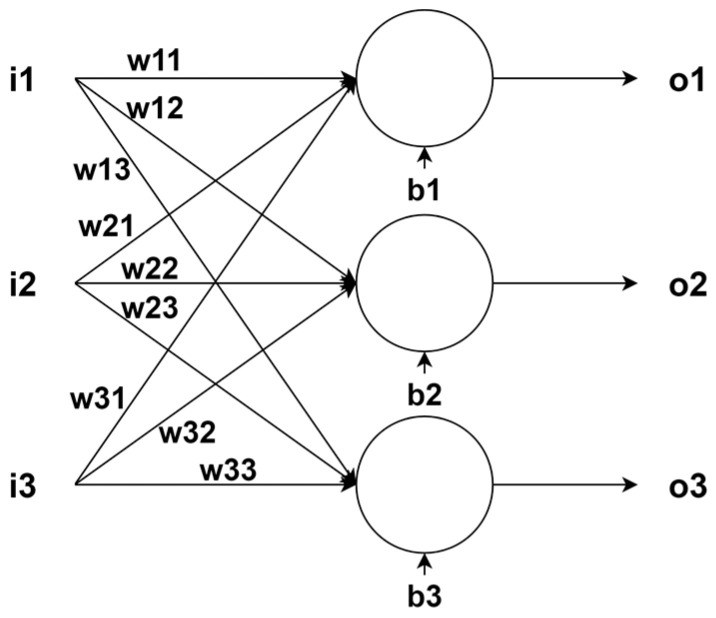
Illustration of a neural network.

**Figure 7 sensors-24-07525-f007:**
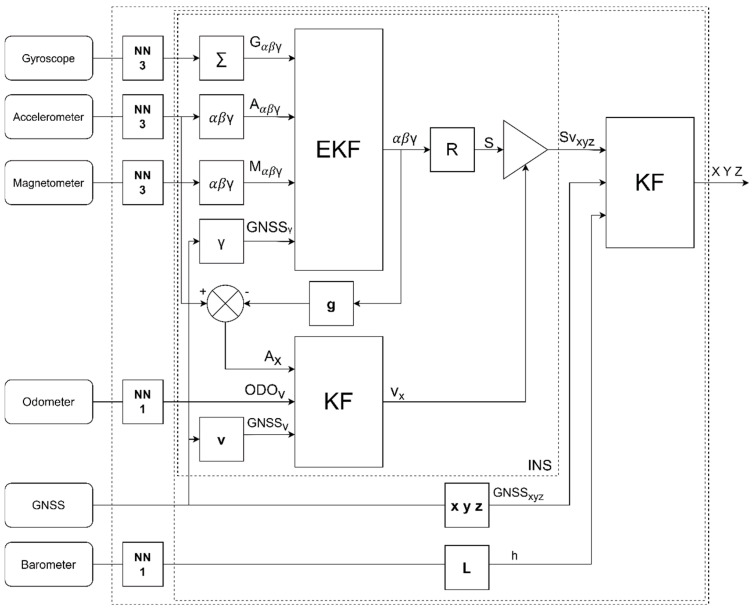
Fusion of input data through neural networks.

**Figure 8 sensors-24-07525-f008:**
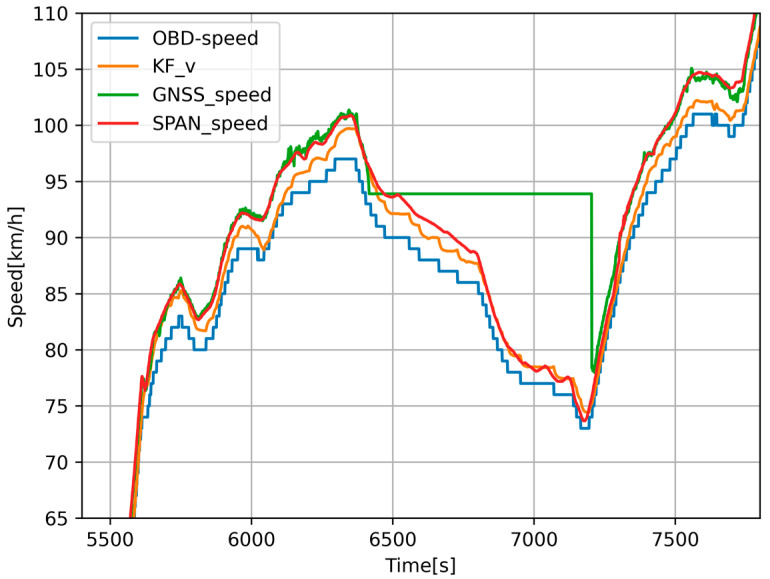
Comparison of estimated speed from odometry (blue), low-cost GNSS (green), navigation-grade GNSS (red), and Kalman filter (orange).

**Figure 9 sensors-24-07525-f009:**
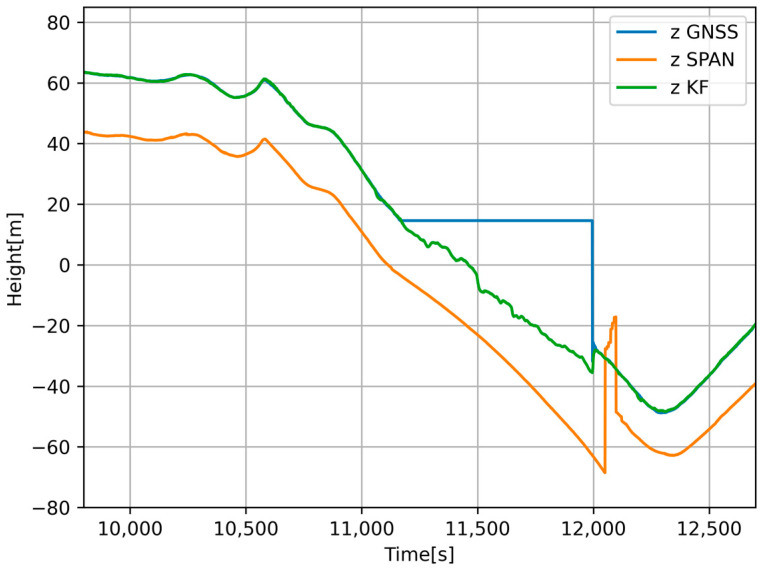
Comparison of estimated relative elevation from low-cost GNSS (blue), navigation-grade GNSS (orange), and Kalman filter (green).

**Figure 10 sensors-24-07525-f010:**
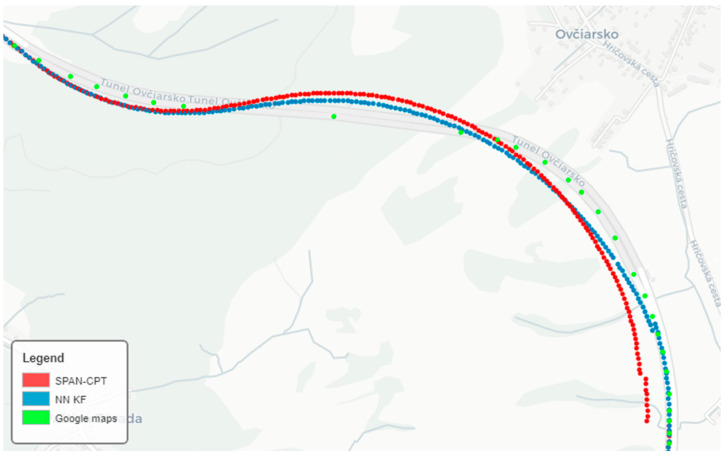
GNSS signal failure in tunnels (red), calibration through error backpropagation (blue).

**Figure 11 sensors-24-07525-f011:**
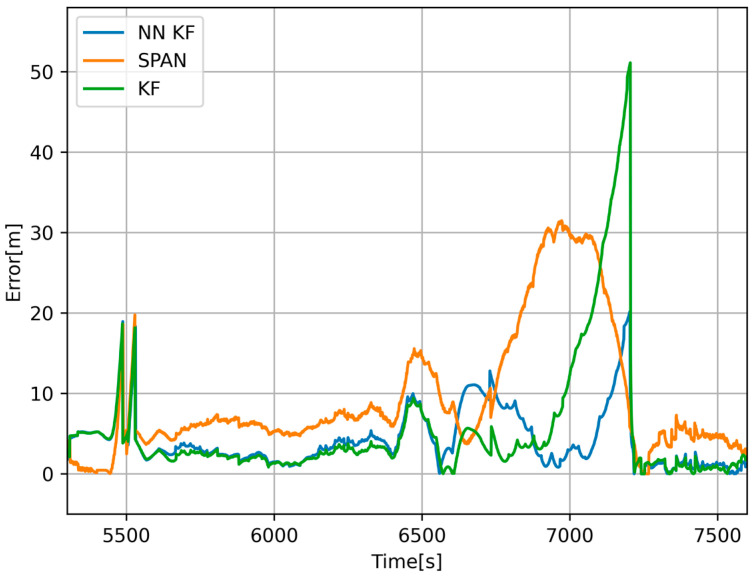
Error of the estimated position from NN-calibrated KF (blue), navigation-grade GNSS+INS (orange), and offline-calibrated KF (green) with respect to the position obtained from map documents while traveling through the first tunnel.

**Figure 12 sensors-24-07525-f012:**
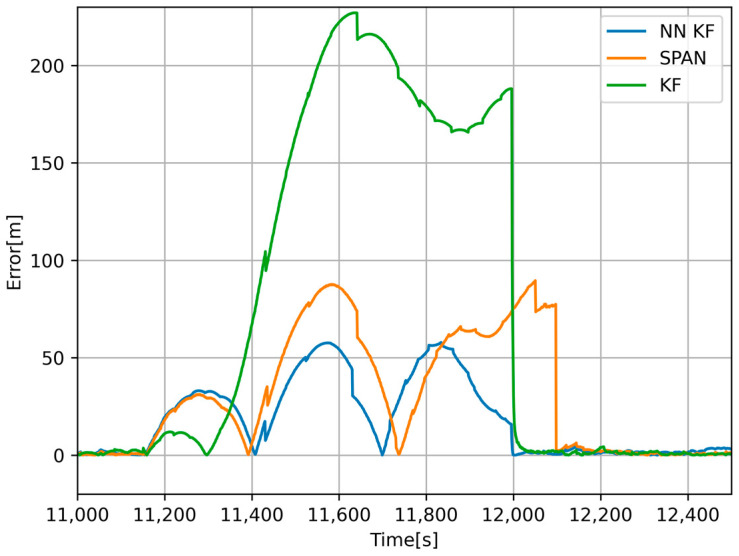
Error of the estimated position from NN-calibrated KF (blue), navigation-grade GNSS+INS (orange), and offline-calibrated KF (green) with respect to the position obtained from map documents while traveling through the first tunnel.

**Table 1 sensors-24-07525-t001:** Comparison of positioning accuracy.

	SPAN-CPT [m]	KF [m]	KF with Backpropagation [m]
Mean average error	6.12	9.19	4.14
RMSE	14.07	34.14	8.92

## Data Availability

Data are available upon request from the corresponding author.

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
