# Peer review of "Online Calibration of Inertial Sensors Based on Error Backpropagation"

_sensors, 2024, doi:10.3390/s24237525_

Round 1
Reviewer 1 Report
Comments and Suggestions for Authors
Several improvements can be suggested to the authors:
1. Some abbreviations are not explained in the paper (KF, EKF, IMU).
2. Line 50: In the Kalman filter theory, the noise covariance matrix is not necessarily constant and can be time-varying (see, e.g., T. Kailath).
3. Line 51: the terms “basic EKF” and “augmented EKF” are not widely used in the Kalman filter literature and thus should be clarified in the paper. Much more common are “open-loop” and “closed-loop” filtering (if the authors actually mean them).
4. Line 56: it is not clear enough what the authors mean by “24/7 applications”. Continuous long-term applications? Hence, it should be clarified what time periods are assumed.
5. Line 90: it is unclear what the “discontinuity between the axes” means. When considering the geometrical errors of inertial sensors, the term “non-orthogonality” (or “misalignment”) is commonly used.
6. Line 188: it should be noted in the paper that Eqs. (3)-(5) are valid only if the misalignment between the odometer frame and the INS body-frame is negligible.
7. Line 198: the phrase in bold should be deleted.
8. Line 206: In Eq. (11), F_k and G_k depend on k, while the right-hand side does not depend on k.
9. Line 213: In Eq. (18), H_k depends on k, while the right-hand side does not depend on k.
10. Line 218: the phrase “dispersion of individual sensor” can lead to misunderstanding by the reader. A much more common variant is “the sensor noise variance”.
11. Line 232: in Eq. (22), y should be replaced by “y_k”.
12. Line 244: the variable “z” means the z-coordinate and error vector at the same time, which may confuse the reader.
13. Line 281: the reference should be added to “Adam algorithm”.
14. Line 295-297: the curves on the plots should be explained in the figure captions.
15. Line 302-304: the authors state that “the maps of the tunnels did not correspond to reality” and the maps of Google are the most accurate. It is not clear how the accuracy of maps was evaluated. A clarification would be highly desirable.
16. Line 307: “tab. 1” should be replaced by “Table 1”.
17. Line 310: the phrase “we compared traditional calibration methods with the data” is not clear. One method can be compared with another method, but not with the data. The term “traditional calibration” should be explained.
18. Line 313: The phrase is not completed.
19. Lines 320-323: the figure captions contain inaccuracy in wording (English language should be checked).
20. Line 330: the mention of the airplane (as a possible carrier) in the paper is not necessary as it comes without further explanation. Therefore, it should either be removed or supplemented with more detailed information.
21. Line 331: it is unclear why the algorithm can also work in the online mode. Has it been tested? An explanation would be desirable.
22. All figure captions should be checked and corrected in accordance with the rules of the journal.
References:
1. Kailath T., Sayed A.H., Hassibi B. Linear estimation. Prentice Hall, Englewood Cliffs. 2000. 854 p.
Author Response
Comments 1: Some abbreviations are not explained in the paper (KF, EKF, IMU).
Response 1: Abbreviations have been explained at the first occurence in the text.
Comments 2: Line 50: In the Kalman filter theory, the noise covariance matrix is not necessarily constant and can be time-varying (see, e.g., T. Kailath).
Response 2: The statement at line 50 has been rewritten into "vanilla version of EKF assumes the noise covariance matrices need to be are constant and apriori known. For time-variant systems, this issue can be ad-dressed by adaptive EKF, which estimates the covariance of inputs from previous samples"...
Comments 3: Line 51: the terms “basic EKF” and “augmented EKF” are not widely used in the Kalman filter literature and thus should be clarified in the paper. Much more common are “open-loop” and “closed-loop” filtering (if the authors actually mean them).
Response 3: we have replaced the "basic EKF" with "vanilla EKF", which is more common. We have added reference to augmented EKF example.
Comments 4: Line 56: it is not clear enough what the authors mean by “24/7 applications”. Continuous long-term applications? Hence, it should be clarified what time periods are assumed.
Response 4: The term 24/7 has been replaced by "continous long-term" to avoid confusion. Thank you for the suggestion.
Comments 5: it is unclear what the “discontinuity between the axes” means. When considering the geometrical errors of inertial sensors, the term “non-orthogonality” (or “misalignment”) is commonly used.
Response 5: We the term "discontinuity" have been replaced with "misalignment" to avoid confusion. Thank you for the suggestion.
Comments 6: Line 188: it should be noted in the paper that Eqs. (3)-(5) are valid only if the misalignment between the odometer frame and the INS body-frame is negligible.
Response 6: We have added a clarification below the equations. Indeed, we assume that the INS axes are aligned with, but the neural networks calibrating each sensor also compensate the misalignment.
Comments 7: Line 198: the phrase in bold should be deleted.
Response 7: The broken reference to Figure 4 has been repaired.
Comments 8: Line 206: In Eq. (11), F_k and G_k depend on k, while the right-hand side does not depend on k.
Response 8: We have replaced our oversimplified notation by correct one, explaining the partial derivative of function f is indeed in the point x_k, u_k.
Comments 9: Line 213: In Eq. (18), H_k depends on k, while the right-hand side does not depend on k.
Response 9: We have replaced our oversimplified notation by correct one, explaining the partial derivative of function h is indeed in the point x_k.
Comments 10: Line 218: the phrase “dispersion of individual sensor” can lead to misunderstanding by the reader. A much more common variant is “the sensor noise variance”.
Response 10: Thanks for the suggestion, the term "dispersion" has been replaced by "noise variance" in all occurences.
Comments 11: Line 232: in Eq. (22), y should be replaced by “y_k”.
Response 11: Corrected, thank you.
Comments 12: Line 244: the variable “z” means the z-coordinate and error vector at the same time, which may confuse the reader.
Response 12: We have revised all EKF symbols. The EKF state has been renamed from x to s, measured output is renamed from y to q and output error has been renamed from z to e. We believe it will avoid all confusion with x, y, z coordinates.
Comments 13: Line 281: the reference should be added to “Adam algorithm”.
Response 13: The reference to original paper has been added.
Comments 14: Line 295-297: the curves on the plots should be explained in the figure captions.
Response 14: The captions have been completely rewritten to clarify the source of data.
Comments 15: Line 302-304: the authors state that “the maps of the tunnels did not correspond to reality” and the maps of Google are the most accurate. It is not clear how the accuracy of maps was evaluated. A clarification would be highly desirable.
Response 15: We have compared the online maps from various sources with official maps from Slovak Road Administration and chosen the best matching online map source.
Comments 16: Line 307: “tab. 1” should be replaced by “Table 1”.
Response 16: The reference to the table has been corrected.
Comments 17: Line 310: the phrase “we compared traditional calibration methods with the data” is not clear. One method can be compared with another method, but not with the data. The term “traditional calibration” should be explained.
Response 17: Thanks for your comment, the sentence was very confusing. We have compared the errors of the proposed neural-calibrated Kalman filter system with the same system, where the neural networks were replaced by standard calibration parameters (gain, bias). The explanation has been added to the paper.
Comments 18: Line 313: The phrase is not completed.
Response 18: The sentence has been rewritten into: "We have calibrated the gyroscope through the ZUPT algorithm (bias calibration was performed whenever the vehicle was stationary)."
Comments 19: Lines 320-323: the figure captions contain inaccuracy in wording (English language should be checked).
Response 19: Captions of Figures 11 and 12 have been revised and completely rewritten.
Comments 20: Line 330: the mention of the airplane (as a possible carrier) in the paper is not necessary as it comes without further explanation. Therefore, it should either be removed or supplemented with more detailed information.
Response 20: The modification of the algorithm for different types of vehicles (aerial, marine...) will be the subject for the further research. We have removed the mention from this article to avoid confusion.
Comments 21: Line 331: it is unclear why the algorithm can also work in the online mode. Has it been tested? An explanation would be desirable.
Response 21: The algorithm processes data sample-by-sample and does not need to know the future values for the correct estimation. That allows its deployment in real-time applications. We have added the explanation into the article.
Comments 22: All figure captions should be checked and corrected in accordance with the rules of the journal.
Response 22: All captions have been re-formatted according to the journal template. Thank you again for all your comments and suggestions, they were very helpful and constructive.
Reviewer 2 Report
Comments and Suggestions for Authors
It is attached in attachment named "comment"

Comments on the Quality of English Language
This paper is well organized.
Author Response
Comments 1: ESKF has better performance than EKF, so why not use ESKF? Please explain.
Response 1: Thanks for your suggestion. Error-state Kalman filter (or even better error-state extended Kalman filter, ESEKF) has the potential to provide slightly better performance and stability for highly non-linear systems compared to vanilla EKF. Due to the characteristics of a passenger vehicle (lower acceleration, low trajectory curvature), we have observed no issues related to the stability of the EKF during the experiments, hence vanilla EKF is sufficient for the application.
Comments 2: More practical experiments should be provided to verify the disadvantages described in Introduction.
Response 2: The mentioned disadvantages of EKF are well known and studied thoroughly in the referenced literature. The aim of the article is not to improve the EKF itself, but rather to improve the readings of the sensors by their calibration.
Comments 3: The format of this paper should be greatly improved.
Response 3: The paper has been thoroughly revised, and format has been corrected according to the journal template.
Comments 4: In the line 198, what ‘Chyba! Nenašiel sa žiaden zdroj 198 odkazov.’.
Response 4: The corrupted link to figure has been corrected.
Comments 5: Many symbols should be properly displayed, for example in line 208, xk,uk. Please revise the similar issues in this paper.
Response 5: The symbols in the paper have been revised and formatted accordingly.
Comments 6: The format of Introduction should be improved.
Response 6: The format of the introduction section has been thoroughly revised, and it has been corrected according to the journal template.
Comments 7: Compared with ESKF, the disadvantages of EKF appling to this paper should be presented.
Response 7: The explanation of the ESKF advantages (better performance at highly non-linear scenarios) has been added to the introduction. Thanks you for all your suggestions and comments.
Round 2
Reviewer 2 Report
Comments and Suggestions for Authors
The authors have revised this paper according to my comments, and I think it can be accepted in present form.